# Structural and Biochemical Insights into Bis(2-hydroxyethyl) Terephthalate Degrading Carboxylesterase Isolated from Psychrotrophic Bacterium *Exiguobacterium antarcticum*

**DOI:** 10.3390/ijms241512022

**Published:** 2023-07-27

**Authors:** Jisub Hwang, Wanki Yoo, Seung Chul Shin, Kyeong Kyu Kim, Han-Woo Kim, Hackwon Do, Jun Hyuck Lee

**Affiliations:** 1Research Unit of Cryogenic Novel Material, Korea Polar Research Institute, Incheon 21990, Republic of Korea; hjsub9696@kopri.re.kr; 2Department of Polar Sciences, University of Science and Technology, Incheon 21990, Republic of Korea; 3Department of Chemistry, Graduate School of General Studies, Sookmyung Women’s University, Seoul 04310, Republic of Korea; vlqkshqk61@skku.edu; 4Department of Precision Medicine, Graduate School of Basic Medical Science (GSBMS), Sungkyunkwan University School of Medicine, Suwon 16419, Republic of Korea; kyeongkyu@skku.edu (K.K.K.); hwkim@kopri.re.kr (H.-W.K.); 5Division of Life Sciences, Korea Polar Research Institute, Incheon 21990, Republic of Korea; ssc@kopri.re.kr

**Keywords:** carboxylesterases, *Exiguobacterium antarcticum*, biocatalyst

## Abstract

This study aimed to elucidate the crystal structure and biochemically characterize the carboxylesterase *Ea*Est2, a thermotolerant biocatalyst derived from *Exiguobacterium antarcticum*, a psychrotrophic bacterium. Sequence and phylogenetic analyses showed that *Ea*Est2 belongs to the Family XIII group of carboxylesterases. *Ea*Est2 has a broad range of substrate specificities for short-chain *p*-nitrophenyl (*p*NP) esters, 1-naphthyl acetate (1-NA), and 1-naphthyl butyrate (1-NB). Its optimal pH is 7.0, losing its enzymatic activity at temperatures above 50 °C. *Ea*Est2 showed degradation activity toward bis(2-hydroxyethyl) terephthalate (BHET), a polyethylene terephthalate degradation intermediate. We determined the crystal structure of *Ea*Est2 at a 1.74 Å resolution in the ligand-free form to investigate BHET degradation at a molecular level. Finally, the biochemical stability and immobilization of a crosslinked enzyme aggregate (CLEA) were assessed to examine its potential for industrial application. Overall, the structural and biochemical characterization of *Ea*Est2 demonstrates its industrial potency as a biocatalyst.

## 1. Introduction

Carboxylesterases (EC 3.1.1.1) are ubiquitous enzymes found in a wide range of organisms, including some viruses [1]. The commonly known mechanism of carboxylesterases is the formation of an alcohol and a carboxylate resulting from the reaction between a carboxylic ester with water molecules. To date, bacterial carboxylesterases have been identified and characterized from various sources and metagenomic samples [2,3]. Thus, the classification and grouping of bacterial carboxylesterases have been continuously changed and updated. Recently, Hitch and Clavel [4] suggested a new classification method for bacterial lipolytic enzymes using 35 families and 11 true lipase subfamilies based on their sequences and conserved motifs. This updated classification system and representative sequences for each group were used for sequence analysis and grouping of new target esterases.

*Exiguobacterium antarcticum* is a psychrotrophic bacterium isolated from microbial mats in Antarctica. The complete genome sequence of *E. antarcticum* has been deposited in the National Center for Biotechnology Information database (GenBank: AY186197.1) [5]. Among the extremophilic carboxylesterases, several enzymes from *Geobacillus stearothermophilus* have shown distinctive thermophilic characteristics in structural and enzymatic analyses [6,7]. However, biocatalysts derived from psychrotrophs and psychrophiles have not been studied as extensively as those derived from thermophiles. Thus, the newly discovered carboxylesterase from *E. antarcticum* is expected to have unique features relative to other extremophilic carboxylesterases in terms of survival mechanisms in cold environments.

In recent studies, several enzymes of the carboxylesterase family have attracted considerable attention because of their ability to degrade polyethylene terephthalate (PET) and PET degradation intermediates [8,9,10]. Carboxylesterases (EC 3.1.1.1) and PETase (EC 3.1.1.101) belong to a subfamily of the carboxyl ester hydrolase family (EC 3.1.1). As they share the canonical α/β hydrolase-fold and conserved catalytic triad (Asp-Ser-His) in their active site [8,11,12], carboxylesterase family enzymes are considered excellent candidates for PET biodegradations.

Recent studies have shown that several carboxylesterases exhibit hydrolytic activities toward bis(2-hydroxyethyl) terephthalate (BHET). For example, the carboxylesterase *Tf*Ca from *Thermobifida fusca* was studied as a PET depolymerization enzyme in 2005 [13]. Even though *Tf*Ca possesses the canonical α/β hydrolase-fold, as well as a highly conserved catalytic triad (Ser-Glu-His), which characterizes the carboxylesterase family, it exhibits additional hydrolytic activity toward PET. In a recent study, *Tf*Ca was confirmed to have depolymerization activity against PET intermediates, BHET, and mono(2-hydroxyethyl) terephthalate (MHET), and its apo- and ligand-binding crystal structures were verified [8]. Additionally, it has been demonstrated that cooperative use of polyester hydrolase together with depolymerization enzymes for PET intermediates, such as BHET and MHET, enhances the enzymatic degradation of PET [14,15]. Thus, it is important to develop biocatalyst candidates for the hydrolysis of PET or PET intermediates that can be applied to industrial-scale biodegradation.

In this study, we identified a new carboxylesterase homolog gene (*Ea*Est2) derived from the psychrotrophic bacterium *E. antarcticum* (GenBank: WP_014971149.1). We compared the amino acid sequence of *Ea*Est2 with those of other bacterial lipolytic enzymes using a recently updated classification method [4]. Moreover, crystal structure determination and biochemical characterization of *Ea*Est2 were performed. Notably, we found that *Ea*Est2 had significant hydrolytic activity toward the BHET substrate. As a result, we investigated the possible interaction mode between *Ea*Est2 and BHET using computational docking simulations. These results suggested the possibility of identifying new BHET degradation enzymes from the esterase family and applying them to industrial-scale biodegradation.

## 2. Results and Discussion

### 2.1. Bioinformatic Analysis of EaEst2

Carboxylesterases are a ubiquitous protein family found in all organisms, including some viruses. *Ea*Est2 originates from the psychrotrophic bacterium *E. antarcticum* and is assumed to have unique structural features relative to carboxylesterases from mesophiles and thermophiles. Comparative sequence analysis was performed on the *Ea*Est2 sequence and other representative sequences for each family of bacterial lipolytic enzymes using a phylogenetic tree (Appendix A). Based on the phylogenetic analysis, the *Ea*Est2 protein can be classified into carboxylesterase Family XIII because it shows a high sequence similarity of 65.04% with Est30 (GenBank ID: AAN81911.1), which is in the same clade. Est30 is a thermostable carboxylesterase derived from *G. stearothermophilus*, and its crystal structure (PDB code 1TQH) was determined in 2004 [6,7]. Additionally, our sequence analysis revealed that *Ea*Est2 possesses a conserved pentapeptide, GLSLG, in its amino acid sequence from 91–95, including a catalytic Ser residue, which is a typical consensus motif in carboxylesterase Family XIII enzymes [16] (Figure 1).

### 2.2. Biochemical Characterization of EaEst2

The substrate specificity of recombinant *Ea*Est2 was examined using various ester compounds and other possible substrates. First, the relative preference of *Ea*Est2 for different acyl chain lengths showed that *Ea*Est2 had the strongest activity toward *p*-nitrophenyl acetate (C2), which had the shortest chain length among the *p*-nitrophenyl esters tested (Figure 2A). Activity gradually decreased as the length of the substrate increased. Additionally, *Ea*Est2 was thought to have no activity against the substrate, which was longer than *p*-nitrophenyl dodecanoate (C12). Among the naphthyl derivatives, *Ea*Est2 showed strong regioselectivity for 1-naphthyl ester substrates. *Ea*Est2 activity was much higher toward 1-naphthyl acetate (1-NA) and 1-naphthyl butyrate (1-NB) than that toward 2-naphthyl acetate (2-NA) (Figure 2B). The optimal pH of *Ea*Est2 was investigated over a pH range from 3.0 to 10.0. *Ea*Est2 showed its maximal activity at pH 7.0, whereas only ~40% of its maximal activity was retained at pH 8.0 (Figure 2C). Under optimal conditions, the enzymatic activity of *Ea*Est2 showed a hyperbolic curve for Michaelis–Menten kinetics, with kinetic parameters (V_max_, K_m_, and kcat/K_m_) determined using *p*-NA. V_max_ and K_m_ values of 5.59 μM s^−1^ and 0.59 mM were obtained, respectively (Figure 2D).

For the chemical stability of *Ea*Est2, the results showed that *Ea*Est2 was highly activated in the presence of 10 and 30% ethanol, with approximately 180 and 160% of the initial activity, respectively. However, the enzyme activity of *Ea*Est2 was almost completely lost in the presence of other chemicals such as sodium dodecyl sulfate (SDS), Tween 20, Triton X-100, isopropyl alcohol (Iso-PrOH), and urea (Figure 3A). Compared with previously studied esterases from the same species (*E. antarcticum*), *Ea*Est2 showed a greater ethanol tolerance of up to 30% and potential for industrial application [17]. In addition, the activity of *Ea*Est2 was highly stable up to 37 °C after 1 h incubation (Figure 3B). However, *Ea*Est2 activity was completely lost within 15 min at 50 and 60 °C. The circular dichroism (CD) results supported the thermal stability of *Ea*Est2, as shown by its T_m_ value, implying that 50% of the secondary structure of *Ea*Est2 was denatured at 52 °C (Figure 3C,D).

We subsequently used a pH-dependent colorimetric assay to examine the hydrolytic properties of *Ea*Est2 with respect to carbohydrates and lipids. Significant *Ea*Est2 hydrolytic activity was detected only against mannose pentaacetate (Man-Ac5) and glyceryl tributyrate (GTB), and *Ea*Est2 also showed weak activity against glucose pentaacetate (Glu-Ac5) and galactose pentaacetate (Gal-Ac5) (Figure 4A,B). However, no activity was observed against other compounds. We also conducted an *Ea*Est2 enantioselectivity analysis using a pH shift assay with (*R*)- and (*S*)-Roche esters (methyl-3-hydroxy-2-methylpropionate). As shown in Figure 4C,D, a color change to yellow was observed only in the enzyme mixture containing the (*S*)-Roche ester, indicating that *Ea*Est2 prefers the (*S*)-enantiomer of the chiral ester to the (*R*)-enantiomer. These results were confirmed by measuring absorbance spectra. Collectively, our biochemical studies indicated that *Ea*Est2 has promising hydrolytic properties and stability and can be used for diverse industrial applications.

### 2.3. Overall Structure of EaEst2

The crystal structure of EaEst2 has been determined at a 1.74 Å resolution in the ligand-free form (Figure 5A). The structure was refined to an *R*_work_ of 19.6% and an *R*_free_ of 21.3%. The asymmetric unit of the crystal (space group *P*2_1_2_1_2_1_) contained one *Ea*Est2 molecule and 169 water molecules. The overall structure of *Ea*Est2 comprised seven β-strands and 10 α-helices and adopted a classical α/β hydrolase fold in the core domain, which was a twisted β sheet surrounded by α-helices and a cap domain (Figure 5B). Three putative catalytic residues, S93, D190, and H220, were located on the β4-α4, β6-α8, and β7-α9 loops, respectively. Notably, the substrate binding pocket of EaEst2 was covered by a cap domain comprising α2-, α5-, and α6-helices, forming a long and narrow shape (Figure 5C).

For molecular characterization, *Ea*Est2 was determined to be approximately 30 kDa by sodium dodecyl sulfate–polyacrylamide gel electrophoresis (SDS-PAGE) and mass spectrometry (Figure 6A,B). Unlike most carboxylesterases, which are thought to exist as dimers in solution, the functional unit of *Ea*Est2 was determined to be a monomer by size-exclusion chromatography [18] (Figure 6C,D). The molecular weight calculated from the correlation equation was approximately 28.46 kDa, and the sequence-based computed molecular weight was 27.69 kDa.

Structural homolog search using the DALI server showed that *Ea*Est2 had the highest structural similarity (Z-core of 41.8) to thermophilic carboxylesterase Est30 from *G. stearothermophilus* (PDB code 1TQH [7]), which has hydrolysis activity even at 70 °C. In addition, the carboxylesterase from *Bacillus stearothermophilus* (PDB code 1R1D), lipase from the goat rumen metagenome (PDB code 4DIU), and monoglyceride lipase from *Bacillus* sp. H257 (PDB codes 4KE6 [19] and 3RLI [20]) showed significant structural similarities to the *Ea*Est2 structure (Table 1). Structural information can also be used for further protein engineering of *Ea*Est2. In addition, a structural comparison of *Ea*Est2 and Est30 may provide useful insights into their different temperature-dependent activities and stabilities.

### 2.4. BHET Hydrolysis Activity of EaEst2

Notably, we found that *Ea*Est2 had strong activity in BHET hydrolysis. HPLC analysis showed a decrease in peak height for BHET after enzymatic reaction with *Ea*Est2. The newly generated peak at 3.17 min of retention time was considered MHET, regarded as a degradation intermediate since the polarity of the eluted compound was higher than that for BHET and lower than that for the TPA molecule (Figure 7). In addition, subsequent MHET degradation by *Ea*Est2 was not observed. Collectively, the *Ea*Est2 enzyme can degrade BHET into MHET by cleavage of the ester bond and cannot utilize MHET as a substrate. Previous studies showed that MHET released during PET degradation is a possible PET degrading enzyme inhibitor [14,22,23,24]. This implied that MHET-degrading enzymes are needed to complete the decomposition of PET into TPA and ethylene glycol (EG). Thus, it is possible to utilize *Ea*Est2 enzymes in PET degradation research or industries that need to degrade BHET specifically.

### 2.5. Immobilization of EaEst2

Enzyme immobilization is an effective strategy to improve the stability and recyclability of free enzymes. The immobilization of *Ea*Est2 has been characterized for biotechnological and industrial applications. *Ea*Est2 was immobilized as a crosslinked enzyme aggregate (CLEA) by precipitation with ammonium sulfate and crosslinking with glutaraldehyde. Scanning electron microscopy (SEM) images of the CLEAs show the morphological structure of the amorphous aggregate of *Ea*Est2 (Figure 8A). Figure 8B shows the reusability of the immobilized *Ea*Est2 after *p*-NA hydrolysis. Immobilized *Ea*Est2 retained more than 100% of its initial activity after nine reutilization cycles. The thermostability of soluble and immobilized *Ea*Est2 was determined by incubation at 50 °C for various intervals. Notably, immobilized *Ea*Est2 exhibited significantly enhanced activity and stability. Approximately −70% of its initial activity was retained after exposure to 50 °C for 60 min. However, all of its soluble enzymatic activity was lost after only 15 min (Figure 8C). In addition, the chemical stability of the immobilized *Ea*Est2 was higher than that of the soluble enzyme, especially in 30% EtOH conditions (Figure 8D). These results suggest that immobilized *Ea*Est2 can be effectively reutilized for potential industrial applications.

## 3. Materials and Methods

### 3.1. Expression and Purification of Recombinant EaEst2

The gene encoding carboxylesterase in the *E. antarcticum* genome was cloned into the pET-28a vector using *Nhe*I and *Xho*I restriction enzymes and then heterologously expressed in *Escherichia coli* BL21 (DE3) cells with the 6×-histidine tag at the N-terminal of the *Ea*Est2 amino acid sequence. The recombinant *E. coli* cells containing *Ea*Est2 were cultured in Luria–Bertani (LB) medium with kanamycin (50 μg/mL) to an OD_600_ of 0.6 at 37 °C. Subsequent culture was performed by adding 1 mM of isopropyl-β-D-1-thiogalactoside (IPTG) for 4 h. Cell pellets were then collected via centrifugation at 2000× *g* rpm for 15 min, after which the cells were disrupted by sonication in a cell lysis buffer (5 mM imidazole, 20 mM Tris-HCl, 200 mM NaCl, pH 7.5). Thereafter, the supernatant was separated by centrifugation at 16,000× *g* rpm for 40 min and subsequently loaded onto a HisTrap column (GE Healthcare, Chalfont Saint Giles, UK) for purification. The recombinant *Ea*Est2 protein was subsequently eluted using a high-concentration imidazole buffer (300 mM imidazole, 20 mM Tris-HCl, 200 mM NaCl, pH 7.5), and the pooled sections were transferred into the final buffer (20 mM Tris-HCl, 200 mM NaCl, pH 7.5).

### 3.2. Hydrolase Activity

For the esterase activity of *Ea*Est2, *p*-nitrophenyl (*p*-NP) esters with different acyl chain lengths, including *p*-nitrophenyl acetate (C2, *p-*NA), butyrate (C4, *p-*NB), hexanoate (C6, *p-*NH), octanoate (C8, *p-*NO), decanoate (C10, *p-*NDec), and dodecanoate (C12, *p-*NDo), were used as substrates. For the enzymatic reaction, 10 μg of *Ea*Est2 and 0.25 mM *p*-NP esters were used. The release of *p*-nitrophenol was measured at 405 nm using an EPOCH2 microplate reader (BioTek, Winooski, VT, USA). The regioselectivity of *Ea*Est2 was also studied using 0.05 mM 1-naphthyl acetate (1-NA, α-naphthyl acetate), 2-naphthyl acetate (2-NA, β-naphthyl acetate), and 1-naphthyl butyrate (1-NB, α-naphthyl butyrate) as substrates. Absorbance was measured at 310 nm. All experiments were performed in triplicate, and the highest activity against *p*-NA (C2) and 1-naphthyl acetate (1-NA) was defined as 100% of relative activity.

A pH shift colorimetric assay was performed to evaluate the carboxylesterase activity and enantioselectivity of *Ea*Est2. Phenol red solution was used as a pH indicator, and hydrolytic activity was detected based on the color and changes in absorbance. To determine carboxylesterase activity, carbohydrate esters (glucose pentaacetate (Glu-Ac5), mannose pentaacetate (Man-Ac5), galactose pentaacetate (Gal-Ac5), cellulose acetate (CA), and N-acetyl-glucosamine (*N-*Glu-Ac)), lipid (glyceryl tributyrate (GTB), glyceryl trioleate (GTO), fish oil (FO), and olive oil (OO)) were used as *Ea*Est2 substrates. The enantioselectivity of *Ea*Est2 was examined using (*R*)- and (*S*)-Roche esters (methyl-3-hydroxy-2-methylpropionate) in a pH shift-colorimetric assay and by scanning the absorbance at 300–600 nm using UV/vis spectra.

### 3.3. Optimal pH and Chemical Stability Assay

The optimal pH was investigated using *p-*NA (C2) as a substrate in a reaction mixture containing 0.25 mM *p*-NA (C2) and 10 μg of *Ea*Est2. The optimal pH was determined by measuring the enzyme activity of *Ea*Est2 from pH 3.0–10.0 at 25 °C. The following buffers were used: 50 mM citrate-NaOH (pH 3.0–6.0), 100 mM phosphate-NaOH (pH 7.0), 50 mM Tris-HCl (pH 8.0), and 20 mM glycine-NaOH (pH 9.0–10.0).

The chemical stability of *Ea*Est2 was investigated using *p*-NA substrate in buffers containing diverse chemicals. The residual activity of *Ea*Est2 was measured by spectrophotometric assay after incubating the enzyme for 1 h under diverse conditions, including 10 and 30% ethanol (EtOH), 0.2% sodium dodecyl sulfate (SDS), 1% Tween 20 (Tw20), 1% Triton X-100 (Tx-100), 30% isopropanol (I-PrOH), and 5 M urea. The enzymatic activity against *p-*NA (C2) in buffer alone was defined as 100% of relative activity. All experiments were performed in triplicate, and the standard deviations were calculated and presented in the figures.

### 3.4. Size-Exclusion Chromatography (SEC)

To investigate the functional unit in the aqueous state, size-exclusion chromatography (SEC) was performed using a Superdex 200 10/100 GL column (Cytiva, Marlborough, MA, USA). The protein standard mix used to generate standard and correlation graphs included bovine thyroglobulin (approximately 670 kDa), g-globulins from bovine blood (approximately 150 kDa), chicken egg grade VI albumin (approximately 44.3 kDa), and ribonuclease A type I-A from the bovine pancreas (approximately 15 kDa). The protein standard mix and *Ea*Est2 were eluted with 20 mM Tris-HCl (pH 8.0) and 200 mM NaCl buffer at a 0.5 flow rate.

### 3.5. Determination of Thermal Stability

To assess the effect of temperature changes on *Ea*Est2 activity, activity was measured by incubating the enzyme at 0, 15, 25, 37, 50, and 60 °C for 1 h. Each aliquot was obtained every 15 min to measure the residual activity against *p*-nitrophenyl acetate (C2), as described above. The structural denaturation profile was recorded using CD. The secondary structure and thermal unfolding of *Ea*Est2 were evaluated by scanning at 190–280 nm from 5 to 95 °C. Melting temperature (T_m_) was calculated using the molar ellipticity value at 222 nm from the thermal denaturation profile.

### 3.6. Crystallization of EaEst2

Preliminary crystallization screening of *Ea*Est2 was performed in a 96-well plate using the sitting-drop vapor-diffusion method. Commercial crystallization suites MCSG I–IV (Anatrace, Maumee, OH, USA), PGA Screen (Molecular Dimensions, Catcliffe, UK), and a customized suite SGC were used for the initial screening. In the reservoir of each well, 80 μL of crystallization solution was aliquoted, and 400 nL of the purified protein (8 mg/mL) and an equal volume of reservoir solution were mixed in each subwell of the plates using an automatic liquid handling robot (Mosquito; SPT Labtech, Melbourn, UK). The crystal of native *Ea*Est2 was grown using a 0.1 M HEPES: NaOH (pH 7.5) and 20% (*w*/*v*) PEG 8000 mixed solution, and a single crystal appeared after 1-month incubation at 296 K. The obtained crystals were used directly for X-ray diffraction experiments without further optimization.

### 3.7. X-ray Diffraction Data Collection and Structure Determination

A single crystal of *Ea*Est2 was transferred into a cryoprotectant, a mixture of reservoir solution and glycerol, such that the final concentration of glycerol was 25%. The crystal was then placed on a goniometer with a 100 K liquid nitrogen stream at the synchrotron beamline 5C (BL-5C) of the Pohang Accelerator Laboratory (PAL, Pohang, Republic of Korea). Native diffraction data were successfully collected using 360 images rotating at oscillations of 1° per frame on an Eiger 9M detector (Dectris, Baden, Switzerland). HKL-2000 software (HKL Research Inc., Charlottesville, VA, USA) [25] was used for data processing, indexing, integration, and scaling. The crystal structure of *Ea*Est2 was determined to be *P*2_1_2_1_2_1_ with unit cell parameters of a = 50.362 Å, b = 67.789 Å, c = 89.617 Å, and α = β = γ = 90°. In addition, the crystal structure was considered an *Ea*Est2 monomer structure in an asymmetric unit because the Matthews coefficient value was calculated as 2.76 Å^3^ Da^−1^ with 55.5% solvent content [26]. The phasing and generation of electron-density maps were carried out in CCP4 [27] using the program MOLREP [28]. The structure of the carboxylesterase Est30 (PDB code: 1TQH) was used as a template for molecular replacement. Iterative refinement was performed with additional manual model building in Coot [29] and Phenix refinement [30] using the Phenix package [31]. The final model was validated using MolProbity [32] and deposited in the Protein Data Bank (PDB) under the accession code 8HEA. The X-ray diffraction and refinement statistics are presented in Table 2 and all structural figures were generated by PyMOL [33].

### 3.8. BHET Hydrolysis Activity

To verify whether *Ea*Est2 has activity against bis(2-hydroxyethyl) terephthalate (BHET), a 500 mM BHET stock solution was prepared by dissolving it in dimethyl sulfoxide (DMSO). Terephthalic acid (TPA), which is known as one of the final products of BHET, was equally prepared to generate standard HPLC data. The enzyme activity assay was performed in a reaction buffer (100 mM phosphate-NaOH and 100 mM NaCl) at pH 7.5 with 2.5 mM (final concentration) of BHET. The enzyme reaction was initiated by the addition of 100 μg enzyme and incubated at 37 °C for 1 h. The reaction was terminated by adding the same volume of acetonitrile, which was then injected into an Eclipse plus C18 reverse-phase column (150 mm × 4.6 mm, 3.5 μm; Agilent Technologies, Santa Clara, CA, USA) for HPLC analysis.

### 3.9. Immobilization of EaEst2

To prepare the CLEAs of *Ea*Est2, a purified *Ea*Est2 (0.5 mg) was incubated in phosphate buffer (pH 7.5) with 80% ammonium sulfate and 50 mM glutaraldehyde with gentle inverting for 12 h. Thereafter, the suspension was centrifuged at 13,000× *g* rpm at 4 °C for 30 min, and the resulting immobilized *Ea*Est2 was washed thrice with 20 mM Tris-HCl (pH 8.0) and 100 mM NaCl. Immobilized *Ea*Est2 activity was monitored by measuring the hydrolysis of *p*-nitrophenyl acetate (C2, *p*-NA). The thermal stability of immobilized *Ea*Est2 was investigated at 80 °C, and the activity of soluble *Ea*Est2 was set to 100%. To examine its reusability, immobilized *Ea*Est2 was retrieved by simple centrifugation after each reaction. After repeated washing steps (usually thrice), a new substrate was added for another cycle, and the activity of immobilized *Ea*Est2 was measured. For SEM analysis of CLEAs-*Ea*Est2, CLEAs were fixed by 1% osmium tetraoxide in a 50 mM sodium cacodylate buffer (pH 7.2) for 2 h at 4 °C. Fixed CLEAs were then dehydrated using 30, 50, 70, 80, 90, and three times of a 100% ethanol series for 10 min each at 25 °C. Dehydrated CLEAs were then incubated in 100% hexamethyldisilazane for 10 min and further dried in an oven for a minimum of 16 h. The samples were mounted onto metal stubs and sputtered with platinum. SEM analysis was performed using a Carl Zeiss SUPRA 55VP instrument (Carl Zeiss, Oberkochen, Germany).

## 4. Conclusions

In conclusion, these biochemical activity assays revealed the broad substrate specificity of on various molecules containing ester groups such as *p*-nitrophenyl acetate (C2), mannose pentaacetate (Man-Ac5) glyceryl tributyrate (GTB), and (S)-Roche esters. Additionally, for the chemical stability of *Ea*Est2, the increased activity to 180 and 160% of the initial activity was examined with 10 and 30% ethanol, respectively. Notably, *Ea*Est2 showed degradation activity on BHET into MHET as one of the PET decomposition steps. In HPLC analysis, *Ea*Est2 is specifically active on BHET and cannot utilize MHET as a substrate. Furthermore, immobilization and reusability for *Ea*Est2 showed the potential of *Ea*Est2 for industrial application. More specifically, the *Ea*Est2 enzyme can be used to disassemble fine chemicals with ester bonds and decompose environmental pollutants. Collectively, these findings may be useful for the development and modification of new BHET-degrading enzymes using previously known esterase enzymes.

## Figures and Tables

**Figure 1 ijms-24-12022-f001:**
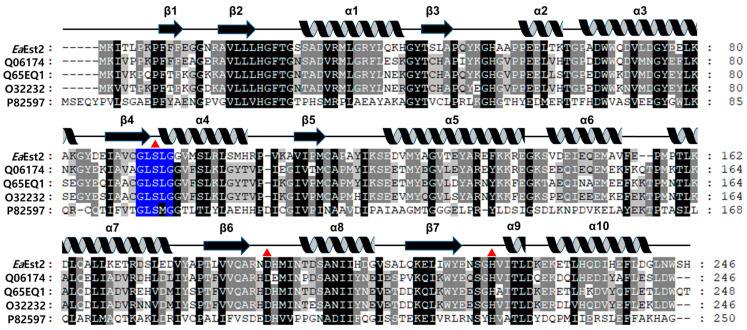
Multiple sequence alignment of *Ea*Est2. Multiple sequence alignment of *Ea*Est2 and selected structural homologs including carboxylesterase from *Geobacillus stearothermophilus* (UniProtKB code Q06174; PDB code 1TQH), esterase from *Bacillus licheniformis* (UniProtKB code Q65EQ1; PDB code 6NKG), carboxylesterase from *Bacillus subtilis,* strain168 (UniProtKB code O32232), and thermostable monoacylglycerol lipase from *Bacillus* sp. (UniProtKB code P82597; PDB codes 3RM3 and 4KEA). The catalytic triad residues (Ser93, Asp190, and His220) are indicated by red triangles, and the consensus motif is colored in blue. Secondary structures obtained from the crystal structure of *Ea*Est2 are presented above the aligned sequences.

**Figure 2 ijms-24-12022-f002:**
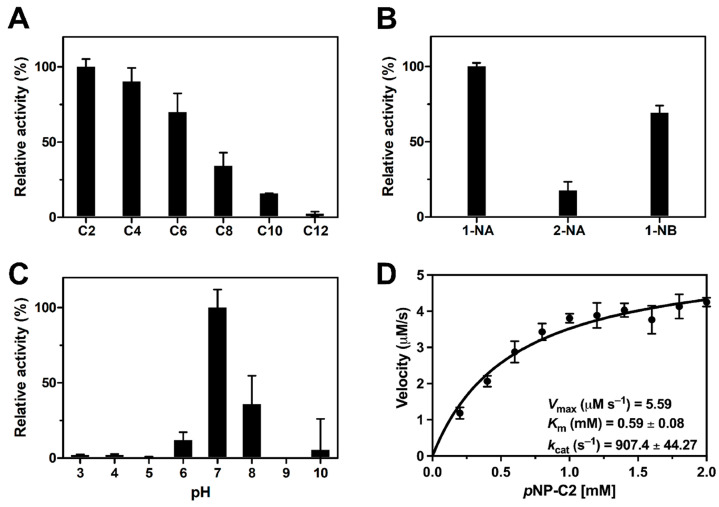
Hydrolase activity assay and optimal pH of *Ea*Est2. (**A**) Esterase activity of *Ea*Est2 using different acyl chain lengths of *p*-nitrophenyl esters from C2–C12. (**B**) Regioselectivity assay of *Ea*Est2 using naphthyl esters as substrates. (**C**) The activity of *Ea*Est2 in different pH buffers (pH 3.0–10.0) using *p*-nitrophenyl acetate (C2) as a substrate. (**D**) Kinetic parameters of *Ea*Est2 toward *p*-nitrophenyl acetate (C2) were obtained by fitting the curve to the Michaelis–Menten equation. All experiments were performed in triplicate.

**Figure 3 ijms-24-12022-f003:**
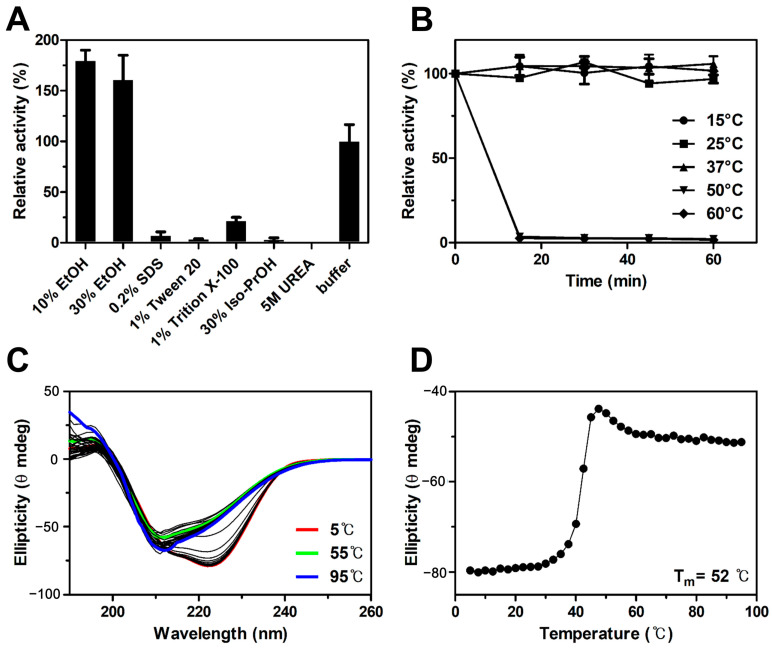
Chemical and thermal stability assay of *Ea*Est2. (**A**) The chemical stability of *Ea*Est2 in various inhibitory chemicals. *Ea*Est2 with buffer alone represented 100% of its relative activity. (**B**) Thermostability of *Ea*Est2. The enzyme was incubated at the indicated temperature, and residual activity was measured at 15 min intervals. (**C**) Thermal denaturation profile of *Ea*Est2 in the range of 5–95 °C. (**D**) Melting temperature (T_m_) of *Ea*Est2 was represented by Boltzman’s equation fitting.

**Figure 4 ijms-24-12022-f004:**
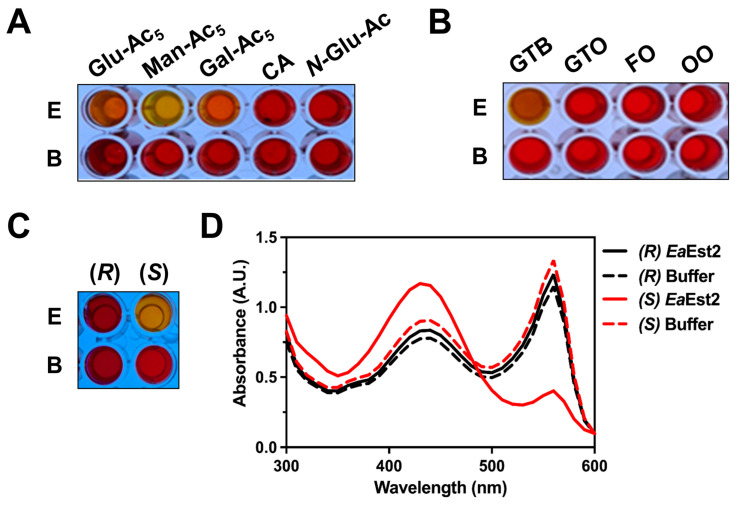
Hydrolysis of various substrates by *Ea*Est2. The hydrolysis of (**A**) carbohydrate esters (glucose pentaacetate (Glu-Ac_5_), mannose pentaacetate (Man-Ac_5_), galactose pentaacetate (Gal-Ac_5_), cellulose acetate (CA), and N-acetyl-glucosamine (*N*-Glu-Ac)) and (**B**) lipids (glyceryl tributyrate (GTB), glyceryl trioleate (GTO), fish oil (FO), and olive oil (OO)) by *Ea*Est2. (**C**) The hydrolysis of (*R*)- and (*S*)-Roche esters by *Ea*Est2. (**D**) UV/vis spectra of samples in (**C**). All colorimetric analyses were performed using phenol red as a pH indicator. All experiments were conducted in triplicate.

**Figure 5 ijms-24-12022-f005:**
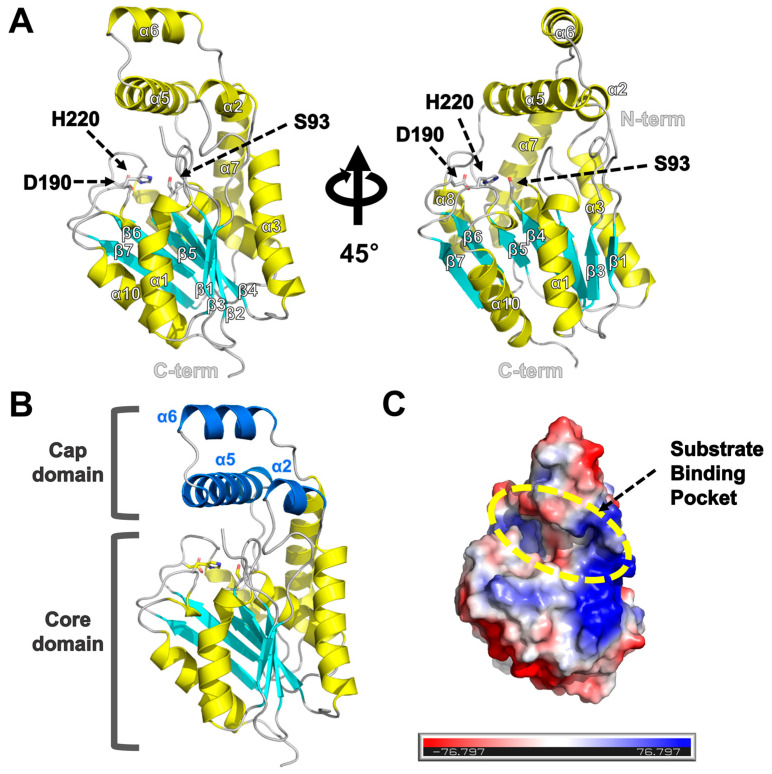
Overall structure of *Ea*Est2. (**A**) Graphic representation of *Ea*Est2 structure. The α-helices and β-strands are colored yellow and cyan, respectively. The conserved catalytic triad of Ser93, Asp190, and His220 is shown using a stick model. (**B**) The cap domain of *Ea*Est2 comprises three α-helices (α2, α5, and α6), which are colored in blue. (**C**) Representation of the electrostatic model of *Ea*Est2; the substrate binding pocket is marked by the yellow dotted circle.

**Figure 6 ijms-24-12022-f006:**
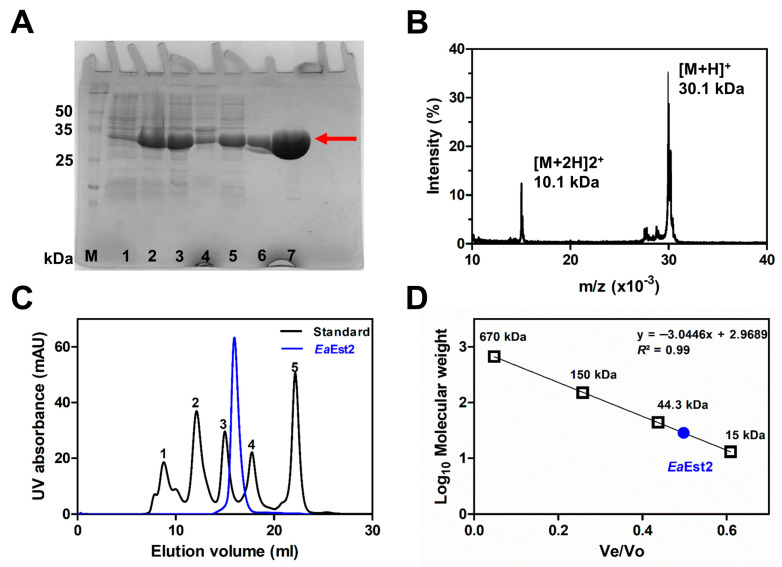
Purification and oligomeric state of *Ea*Est2 in solution. (**A**) SDS-PAGE analysis of samples collected in *Ea*Est2 purification steps. M: marker; 1: before induction; 2: 20 h post-induction by IPTG; 3: supernatant; 4: pellet after sonication and centrifugation; 5: flow-through; 6: wash-through; 7: elution during IMAC purification. The red arrow indicates purified *Ea*Est2. (**B**) Mass spectrometric analysis of *Ea*Est2 (**C**) Size-exclusion chromatography of protein standard mixture (black) and *Ea*Est2 (blue). The standard mixture contains 1: bovine thyroglobulin, 2: γ-globulins, 3: chicken egg albumin grade Ⅵ, 4: ribonuclease A type I-A from bovine pancreas, and 5: *p*-aminobenzoic acid. (**D**) Calculated molecular weight of *Ea*Est2 by linear regression analysis.

**Figure 7 ijms-24-12022-f007:**
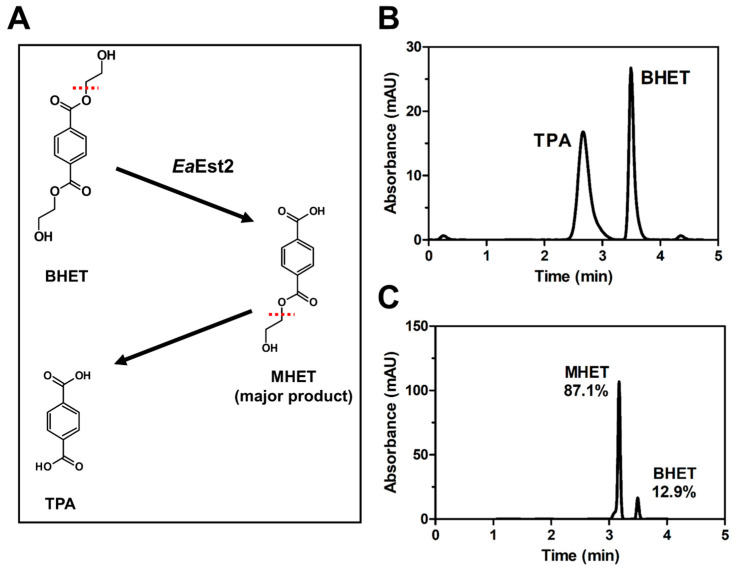
BHET hydrolysis by *Ea*Est2. (**A**) The hydrolysis scheme of BHET by *Ea*Est2. (**B**) HPLC analysis of 1 mM terephthalic acid (TPA) and BHET. (**C**) HPLC analysis of BHET hydrolysis by *Ea*Est2. The relative areas of MHET and BHET peaks are noted.

**Figure 8 ijms-24-12022-f008:**
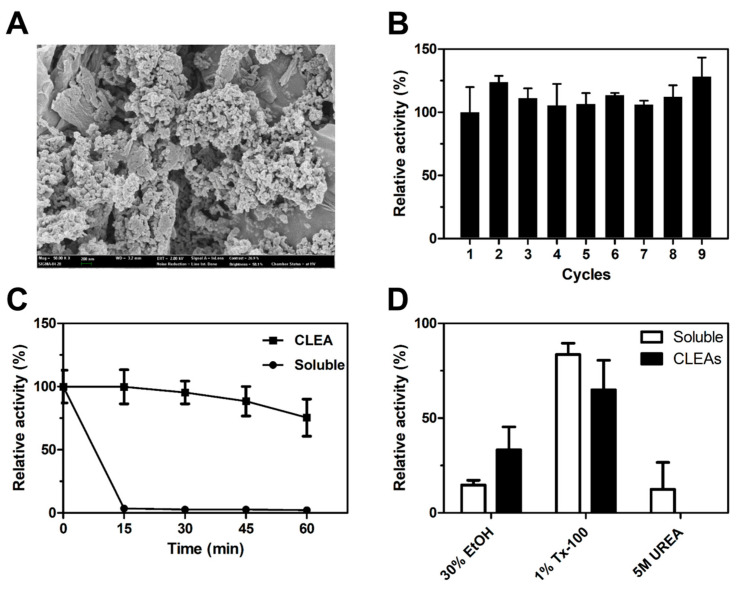
Characterization of crosslinked enzyme aggregates (CLEAs) of *Ea*Est2. (**A**) Field-emission scanning electron microscopic images of CLEAs-*Ea*Est2 at 50,000× (**B**) Reusability of CLEAs-*Ea*Est2 investigated over nine cycles. Comparison of (**C**) thermal and (**D**) chemical stabilities of CLEAs-*Ea*Est2 and soluble *Ea*Est2. In all experiments, *p*NP-C2 is used as a substrate. All experiments were conducted in triplicate.

**Table 1 ijms-24-12022-t001:** Structural homologs of *Ea*Est2 from a DALI search.

Protein	PDB Code	DALI Z-Score	Sequence Identities (%)	Reference
Carboxylesterase Est30 from *Geobacillus stearothermophilus*	1TQH	41.8	65 (240/242)	[7]
Carboxylesterase from *Bacillus stearothermophilus*	1R1D	41.7	63 (240/242)	N/A
Lipase from goat rumen metagenome	6NKG	41.5	60 (243/245)	N/A
Engineered protein from synthetic construct	4DIU	41.5	64 (243/245)	N/A
D196N mutant of monoglyceride lipase from *Bacillus* sp. H257	4KE6	32.7	33 (209/226)	[19]
Esterase D from *Lactobacillus rhamnosus*	3DYI	32.6	28 (231/240)	N/A
Monoacylglycerol lipase from *Bacillus* sp. H257	3RLI	31.8	33 (211/243)	[20]
Monoacylglycerol lipase from thermophilic *Geobacillus* sp. 12AMOR	5XKS	31.6	30 (214/252)	N/A
LipS lipolytic enzyme	4FBL	31.4	28 (211/245)	[21]

**Table 2 ijms-24-12022-t002:** X-ray diffraction data collection and refinement statistics.

Data Collection	*Ea*Est2
X-ray source	PAL 5C
Space group	*P*2_1_2_1_2_1_
Unit-cell parameters (Å, °)	a = 50.36, b = 67.79, c = 89.62 α = 90, β = 90, γ = 90
Wavelength (Å)	0.9796
Resolution (Å)	50–1.74 (1.77–1.74)
Total reflections	420,126
Unique reflections	32,327 (1560)
Average I/σ (I)	53.7 (5.8)
*R*_merge_ ^a^	0.074
Redundancy	13.0 (13.3)
CC (1/2) (%)	99.9 (97.9)
Completeness (%)	100 (99.5)
Refinement	
Resolution range (Å)	50–1.74 (1.79–1.74)
No. of reflections of working set	32,192 (2522)
No. of reflections of test set	1619 (120)
*R*_cryst_ ^b^	0.196 (0.231)
*R*_free_ ^c^	0.213 (0.262)
R.m.s. bond length (Å)	0.016
R.m.s. bond angle (°)	1.78
No. of atoms	
Protein	1931
Solvent	169
Average B value (Å^2^)	
Protein	25.52
Solvent	35.17

^a^ *R*_merge_ = ∑||<I> − I|/∑<I>. ^b^ *R*_cryst_ = ∑||Fo| − |Fc||/∑|Fo|. ^c^ *R*_free_ calculated with 5% of all reflections excluded from refinement stages using high-resolution data. Values in parentheses refer to the highest-resolution shells.

## Data Availability

All data analyzed during this research have been included in this article and Appendix A.

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
