# Peer review of "Structural and Biochemical Insights into Bis(2-hydroxyethyl) Terephthalate Degrading Carboxylesterase Isolated from Psychrotrophic Bacterium Exiguobacterium antarcticum"

_ijms, 2023, doi:10.3390/ijms241512022_

Round 1
Reviewer 1 Report
The article «Structural and biochemical insights into bis(2-hydroxyethyl) terephthalate degrading carboxylesterase isolated from psychrotrophic bacterium Exiguobacterium antarcticum» is attracted to a relevant topic.
This study aimed to elucidate the structure and biochemically characterize the carboxylesterase EaEst2, a thermotolerant biocatalyst derived from Exiguobacterium antarcticum, a psychrotrophic bacterium. EaEst2 was successfully cloned and overexpressed in Escherichia coli. Sequence and phylogenetic analyses showed that EaEst2 belongs to the Family XIII group of carboxylesterases. It is established that EaEst2 has an optimal pH 7.0, losing its enzymatic activity at temperatures above 50 °C. Computational docking simulations were performed to understand how bis(2-hydroxyethyl) terephthalate binds to the active site of EaEst2. Finally, the biochemical stability and immobilization of a crosslinked enzyme aggregate were assessed to examine its potential for industrial application.
The article is well structured, written in sufficient detail and logically. Introduction covers fundamental works in this direction and modern literary sources.
I have only one remark. In my opinion, the conclusion should be expanded, adding key results by specifying key numerical values and materials on the prospects and possible current directions of research in the field.
Reviewer 2 Report
Hwang et al. identified Esterase2 from Exiguobacterium antarcticum (EaEst2) through biochemical experiments and high-resolution crystal structure analysis. Considering the extensive research on esterases/lipases and the availability of crystal structures, the biochemical experiment and crystal structure analysis of EaEst2 conducted by the authors may not be highly specific or novel. However, the finding that EaEst2 is capable of decomposing BHET, which is of significant environmental interest, is likely to be of interest to researchers in related fields. Meanwhile, the authors presented interesting claims regarding BHET, but they did not provide clear experimental evidence to support them. Therefore, I believe that a more detailed study of BHET decomposition is essentially necessary to agree with the authors' claim.
1. The author's report lacks novelty in the general biochemistry and structure description, and there is excessive focus on the properties of esterases, which is not surprising. The significance of esterase activity measured using various substrates, such as pNP, 1-NA, and 1-NB, lacks specificity. I believe that conducting further in-depth biochemistry experiments on BHET degradation activity and comparing it with other BHET-degradable proteins would enhance the quality of this study.
2. The computational docking experiments performed by the authors on BHET/EaEst2 are unprofessional and provide limited information. Moreover, the author did not conduct any supporting experiments, such as mutagenesis, to validate the docking results. Consequently, it is difficult to place trust in the authors' computational docking experiments.
Minor
1. 'EaEst2 was successfully cloned and overexpressed in Escherichia coli.' It is not appropriate for this statement to be in the abstract.
2. Authors should read carefully around the text and correct italics for strains or subscripts for kinetics.
3. The references are duplicated in the Reference section.
Minor editing of English language required
Round 2
Reviewer 2 Report
The authors have addressed the reviewers' concerns and support the publication of the revised manuscript.
Minor editing of English language required